# Synthesis of Biodegradable Polyester–Polyether with Enhanced Hydrophilicity, Thermal Stability, Toughness, and Degradation Rate

**DOI:** 10.3390/polym14224895

**Published:** 2022-11-13

**Authors:** Xuedong Lv, Haitao Lin, Zhengxiang Wang, Ruixue Niu, Yi Liu, Yen Wei, Liuchun Zheng

**Affiliations:** 1School of Textile Science and Engineering, Tiangong University, Tianjin 300387, China; 2China Huanqiu Contracting & Engineering Corp, Beijing 100029, China; 3School of 2011, Nanjing Tech University, Nanjing 211816, China; 4Department of Chemistry, Tsinghua University, Beijing 100084, China

**Keywords:** biodegradable polyester, poly(butylene succinate), hydrolysis degradation, poly(ethylene glycol)

## Abstract

Novel poly(butylene succinate-butylene furandicarboxylate/polyethylene glycol succinate) (PBSF-PEG) was synthesized using two-step transesterification and polycondensation in the melt. There are characterized by intrinsic viscosity, GPC, ^1^H NMR, DSC, TGA, tensile, water absorption tests, and water degradation at different pH. GPC analysis showed that PBSF-PEG had high molecular weight with average molecular weight (*M*_w_) up to 13.68 × 10^4^ g/mol. Tensile tests showed that these polymers possessed good mechanical properties with a tensile strength as high as 30 MPa and elongation at break reaching 1500%. It should be noted that the increase of PEG units improved the toughness of the polyester material. In addition, the introduction of PEG promoted the water degradation properties of PBSF, and the copolymer showed a significantly faster water degradation rate when the PEG unit content was 20%. This suggests that the amount of PEG introduced could be applied to regulate the water degradation rate of the copolymers. Hence, these new polymers have great potential for application as environmentally friendly and sustainable plastic packaging materials.

## 1. Introduction

Up to now, as the economy is rapidly developing and the oil reserves are dwindling, people are paying more and more attention to the utilization and development of renewable resources [1,2,3,4]. In this context, the preparation of bio-based polymer materials from renewable resources is considered to have the dual merits of protecting the environment and saving resources, which has become an important research direction for polymer materials [5].

Biodegradable poly(butylene succinate) (PBS) exhibits relatively high melt temperature, good thermal stability, and processability when compared to other aliphatic polyesters [6,7,8]. Thus, it is considered as a promising alternative to traditional petroleum-based and nondegradable polymers such as polyethylene or polypropylene [9,10]. Recently, new biobased and biodegradable copolyesters have been synthesized using PBS segments in copolymers, while 2,5-furandicarboxylic acid (FDCA) or the 2,5-tetrahydrofurandimethanol (THFDM) unit was used as the segment component [11]. The copolymers have demonstrated excellent flexibility with elongation at break up to 562%, high tensile strength ranging from 30 to 22 MPa, excellent biodegradability, and high biocompatibility. By regulating the ratio of PBS to FDCA, copolyesters can be prepared with a wide range of mechanical properties (from rigid to flexible materials) and degradation rates.

Polyethylene glycol (PEG) is a polymer with excellent properties such as hydrophilicity and biocompatibility. Thus, it has been approved by the U.S. Food and Drug Administration (FDA) for a variety of medical applications [12,13,14]. The availability of various types of PEG in the market offers various possibility [15]. Compared to copolymers synthesized from short PEG segments, the copolymer based on longer PEG segment (a molecular weight of 2000 g/mol) exhibits a higher swelling degree in water and faster in vitro degradation rate [16]. In recent years, various biodegradable aliphatic polyesters and copolyesters such as poly(glycerol sebacate) [17], poly(ε-caprolactone) [18], poly(glycolic acid) [19], and their PEGylated derivatives [19,20,21,22] have been synthesized and their properties have been studied. Zhou synthesized high molecular weight polyester copolymers from succinic acid (SA), 1,4-butanediol (BDO), and PEG through two steps of transesterification and polycondensation using an efficient catalyst [23]. These results showed that all PBS-PEG copolymers showed adjustable melting points (*T*_m_) by varying the content of PBS hard segments. In another study, Wang synthesized a fully shape-memory multiblock polyether-ester (PBS-PEG) consisting of a crystallizable PBS hard segment and PEG soft segment from the polycondensing of SA, BDO, and PEG [24]. The results indicated that PBS-PEG showed microphase separation resulting from its multiblock structure. The results of the tensile tests showed that the PBS-PEG copolymer exhibited excellent shape memory properties when both soft and hard segments were sufficiently crystalline. Through the introduction of hydrophilic PEG, the hydrophilicity of the PBS-PEG copolymers has also been improved. With the introduction of PEG, these biodegradable multiblock copolymers have great potential to be applied as medical device applications due to their excellent shape memory properties and biocompatibility [25,26,27]. Though PEG based copolymers have excellent biocompatible properties, the ether bonds in the PEG chain segments tend to degrade under light exposure, resulting in the deterioration of the mechanical properties. Subsequently, the shelf life of the copolymers are shortened [28,29,30]. Our previous work [31] showed that PBSF could block UV light, so PBSF was introduced into the copolymers to conquer the drawback of the light instability of PEG.

Considering the block flexibility and hydrophilicity can be tailored by a proper choice of building blocks, this study aimed to create novel PBSF-PEG copolymers for the first time by a new two-step process of transesterification and polycondensation. A series of PBSF-PEG were synthesized to systematically investigate the effect of the content of PEG soft segments on the performance of PBSF-PEG. The copolymers were characterized by viscosity, GPC, ^1^H NMR, DSC, TGA, tensile, water absorption tests, and water degradation at different pH.

## 2. Experimental Section

### 2.1. Materials

DMFD was provided by the Ningbo Institute of Materials Technology and Engineering, Chinese Academy of Sciences (Ningbo, Zhejiang, China), DMSu was bought from Yuanli Chemical Co., Ltd. (Weifang, Shandong, China), BDO (99%) was purchased from Mitsubishi Chemical (Japan), titanium(IV) butoxide (TBT), Zn(Ac)_2_ (99%), polyethylene glycol (*M*_n_ = 2000 g/mol), phenol, tetrachloroethane, and chloroform were purchased from Aladdin Reagent Co. (Shanghai, China), All chemicals were used as received.

### 2.2. Synthesis of Copolymers

PBSF-PEG was synthesized through two steps of transesterification and subsequent polycondensation. In short, DMSu, DMFD, and BDO (ester to alcohol molar ratio of 2.2:1) and Zn(Ac)_2_ (0.08 mol%, on the basis total molar of DMFD or DMSu) were added into the kettle under mechanical stirring and N_2_ inlet. Subsequently, it was heated to 150 °C, and Zn(Ac)_2_ was fed into the kettle. The reaction system was heated to 180 °C under the protection of nitrogen gas and kept until the amount of methanol reached the theoretical value. Then, the temperature was increased to 210–230 °C. After that, polyethylene glycol (PEG_2000_) (*M*_n_ = 2000 g/mol) and titanium(IV) butoxide (0.1 *wt%* of total weight of DMFD and DMSu) were added before the pressure was slowly reduced to 20–50 Pa. The polycondensation reaction stopped when the Weissenberg effect emerged, and PBSF or PBSF-PEG were obtained.

### 2.3. *Characterization of* Copolymers

The intrinsic viscosities of the synthesized polymers were measured at 25 °C with an Ubbelohe viscometer (0.792 nm). A total of 0.125 g of the sample was dissolved in a mixed solvent of tetrachloroethane/phenol (50/50 *w*/*w*) to achieve a homogeneous solution. The solutions had a concentration of 5 mg/mL. The intrinsic viscosity was obtained according to Equations (1) and (2):(1)$$\eta_{\mathrm{sp}}=\frac{t-t_{0}}{t_{0}}$$
(2)$$\left[\eta \right]=\frac{\sqrt{1+1.4\eta_{\mathrm{sp}}^{-1}}}{0.7C}$$
where *η*_sp_ is the specific viscosity; *C* is the concentration; *t* and *t*_0_ are the flow time of the solution and pure solvent, respectively.

The molecular weight and distribution of PBSF-PEG were measured with GPC (Agilent 142 PL-GPC220) at 40 °C. The sample concentration was about 1.0 mg/mL. Chloroform was used as the eluent and the flow rate was 1.0 mL/min. The molecular weight was obtained with polystyrene (3070 to 258,000 g/mol) as the standard.

The composition and chemical structures of all polymers were studied by ^1^H NMR spectra on a Bruker AVIII 400 NMR spectrometer at 25 °C. The polymers were dissolved in CDCl_3_, and the internal standard was tetramethysilane (TMS).

FTIR spectra were texted using a Bruker ALPHA spectrometer with attenuated total reflection (ATR) units between 500 and 4000 cm^−1^ at a resolution of 4 cm^−1^. Samples of dried polymer films were scanned 32 times to improve the signal-to-noise ratio.

The thermal properties were investigated with a DSC (Perkin-Elmer Instrument). Samples were first heated from −60 °C to 180 °C at a heating rate of 10 °C min^−1^ and kept at 180 °C for 5 min to eliminate the thermal history. Subsequently, they were cooled to −100 °C at a cooling rate of 10 °C/min and held there for 5 min. Then, the samples were reheated to 180°C with a heating rate of 10 °C/min, and maintained at this temperature for 5 min. Finally, the temperature was reduced to −100 °C at a cooling rate of 10 °C/min. Both heating and cooling curves were recorded. The melting temperature (*T**_m_*), entropy of melting (∆*H_m_*), glass transition temperature (*T*_*g*_), cold crystallization temperature (*T*_*cc*_), and entropy of cold crystallization (∆*H*cc) were obtained.

The crystalline phase content in the hard segments (*X_h_*) was calculated using Equation (3) for the PBS segments and ∆*H_m_* is the enthalpy of melting of the polymer, which is the reference value for 100% crystalline PBS ($\Delta H_{m}^{\theta }$ = 110.3 J/g) [32].
(3)$$X_{c}=W_{h}\%\times \frac{\Delta H_{m}}{\Delta H_{m}^{\theta }}\times 100\%$$
where *W_h_* is the ratio of the hard PBS segment.

Thermal stability was measured by a TGA instrument (NETZSCH TGA/DSC thermogravimetric analysis). TGA was recorded from 40 °C to 800 °C under a dry N_2_ and air atmosphere at a heating rate of 10 °C/min, respectively.

Wide-angle X-ray patterns (WAXD) of the polymers were performed using the wide-angle horizontal goniometer of the BRUKER instrument with CuKα radiation (λ = 0.154 nm) from 10 to 45° and a scanning speed of 2°/min at room temperature. The crystallinity degree (*X_c_*) was calculated using the WAXD fitting software.

The water uptake experiments were carried out in distilled water at 37 °C for 3 days, using dumbbell-shaped dry samples of the polymers (10.0 × 4.0 × 2.0 cm). After 3 days, samples were taken out from water and the surface water was removed using filter paper. The water uptake was calculated according to Equation (4):(4)$$\mathrm{Water}\;\mathrm{uptake}\;\left(\%\right)=\frac{W_{h}-W_{d}}{W_{d}}\times 100\%$$
where *W_d_* the initial weight of dry sample and *W_h_* is the weight of humid sample.

Standard films were tested at 25 °C for the water contact angle. The amount of water droplets was 2 µL and the drop rate was 1 µL/s. The contact angle values of the water droplets of the polyester film were recorded.

The tensile test of the dumbbell-shaped samples (10.0 × 4.0 × 2.0 cm) was performed using an Instron 1122 tensile tester according to ISO 527. The moving speed was 50 mm/min. At least five samples were tested and the average value was reported.

Degradation experiments of the PBSF-PEG films (10.0 × 10.0 × 0.1 mm in size) were carried out in phosphate buffered saline (pH = 6.84) at 37 °C and 0.01 mol/L NaOH solution (pH = 12), respectively, the films were taken out every 3 days, washed with distilled water, and dried under vacuum at 40 °C to constant weight. The media were renewed every 3 days to ensure a stable degradation. The residual weight was calculated by Equation (5):(5)$$\mathrm{Residual}\;\mathrm{weight}\;(\%)=\frac{W_{t}}{W_{0}}\times 100\%$$
where *W*_0_ is the original weight and *W_t_* is the weight after degradation.

## 3. Results and Discussion

PEG was introduced into the PBSF macromolecular chain to improve the hydrophilic properties, water absorption, and tensile properties of the polyester material. We chose transesterification, rather than esterification, and polycondensation to prepare the PBSF-PEG polyester materials, mainly because this reaction does not involve the production of carboxyl groups. The reaction process is shown in Figure 1.

### 3.1. The Chemical Structures and Compositions

The chemical structure of the synthesized copolymers was studied by ^1^H NMR. The results are shown in Figure 2. It can be seen that all PBSF-PEG copolymers had a peak at 7.74 ppm (a), belonging to the signals of H on the furan ring [33]. The peaks at 4.17 ppm (b) and 1.72 ppm (e) are attributed to the BDO residues in the structure of the polymers. There is a slight splitting of the proton peak at this location (b), arising from a different chemical environment of this methylene group. Unfortunately, these peaks cannot be separated, so the degree of randomness (*R*) cannot be calculated. The peak at 3.66 ppm is the H originating from polyethylene glycol. The peaks at 2.63 ppm (d) belong to the methylene peak in the DMSu residue in the structure of the polymers. The final polymer compositions PEG% can be determined from ^1^H NMR according to Equation (6):(6)$$\varphi {\%}_{PEG}=\frac{I_{c}}{I_{c}+I_{e}}\times 100\%$$

The final polymer compositions (PEG wt%) determined from ^1^H NMR are summarized in Table 1. The PEG content in the PBSF-PEG copolyesters can be found to be slightly lower than the feed ratio. It is due to the fact that the low content of –OH end groups of PEG results in the low reaction activity of PEG.

The intrinsic viscosity and molecular weight of the polymers were tested to provide a better insight into the effect of the PEG segment ratios on the polymer properties. The results are concluded in Table 1. The intrinsic viscosities of PBSF and PBSF-PEG ranged from 1.65 to 2.25 dL/g. The values of the weight-averaged molecular weight (*M*_w_) ranged from 10.81 × 10^4^ to 13.68 × 10^4^ g/mol and the polydispersity was between 1.7 and 2.0, which is relatively narrow. Therefore, it can be concluded that high-molecular-weight PBSF-PEG has been successfully synthesized.

### 3.2. FTIR-ATR

The chemical structures of the synthesized polymers were further confirmed by FTIR-ATR (Figure 3). For the PBSF copolymer, the AIR-FTIR displayed furan ring bending vibrations at 748 cm^−1^ and 914 cm^−1^, and furan C=C bond at 1537 cm^−1^. PBSF-PEG copolymer materials showed small bands around 1070 cm^−1^, whose intensity increased with an increasing content of PEG [34]. This band was ascribed to the C–O stretching vibration of the ether bond. Therefore, it could testify to the existence of PEG in the polymer structure [35].

### 3.3. Thermal Properties of Copolymers

The heating and cooling curves are shown in Figure 4 and the corresponding thermal parameters are summarized in Table 2. For PEG2000, there was a cold crystallization peak (32.6 °C) and a melting peak (57.9 °C) in the second heating curve in Figure 4d. However, no cold crystallization peaks and melting peaks of the PEG segment could be observed in PBSF-PEG and PEG. All of the polymers exhibited a melting temperature (*T*m) between 98 °C and 103 °C, suggesting that they are semi-crystalline materials, and the crystallization behavior of the copolymer segment showed a strong dependence on its PEG content. The *X*_c,h_ of PBSF-PEG decreased with an increasing content of the PEG segment [36]. This can be interpreted that with the introduction of the PEG soft segment, the polymer chain flexibility increases, which can be confirmed by the decreased *T*_g_ of the PBSF-PEG copolymer from −26.5 °C to −47.9 °C. This would facilitate the PBSF chains to rearrange into crystalline layers. However, when increasing the PEG content to a certain level, the PEG crystalline domains interfere or limit the crystal growth of PBSF. This is primarily due to the ability of the PEG chain segments to crystallize at room temperature, which in turn affects the overall crystallinity of the copolymer. The resulting crystallinity of the copolymer decreases with the increasing PEG content. As discussed before, the introduction of the PEG soft segment plays a very critical role in the toughening of PBSF. Therefore, the toughening of PBSF polymers can be regulated by the feed ratio.

#### 3.3.1. Thermal Stability of PBSF and PBSF-PEG under N2 and Air

The thermal stability of polyester–polyether materials plays a very critical role in the subsequent use and processing [37]. Generally speaking, the thermal stability of polyether chain segments is poor. Whether PEG would deteriorate the thermal stability of copolymers deserves study. The thermal degradation of PBSF and PBSF-PEG were examined by TGA under a N_2_ atmosphere and air, respectively. Figure 5 shows the TGA and DTG curves of PEG2000, PBSF, and PBSF-PEG under a N_2_ and air atmosphere. Degradation temperatures at 5% weight loss (*T*_,5%_), the temperature of the maximum thermal decomposition rate (*T*_d, max_), and the residual weight at 600°C (*R*_600_%) are summarized in Table 3. It was found that the *T*_5%_ of copolymers by about 10 °C after the introduction of PEG, though the initial decomposition temperature of PEG *T*_d,5%_ was around 265.5 °C, which was much lower than the initial decomposition temperature of PBSF-PEG, ranging from 358 to 370 °C. This is a great improvement as *T*_5%_ decreases for most polyester materials after the introduction of the polyether chain segment. Moreover, similar results were obtained for *T*_d,max_, suggesting that the introduction of PEG chain segments into the PBSF does not decrease the thermal stability of the polyester material, but improves the thermal stability. This is very beneficial for their processing and application. It can be found that under a N_2_ atmosphere, with the increase in the PEG content, *R*_600%_ also has a certain decrease. The enhanced thermal stability can be interpreted that PEG in the nitrogen atmosphere can form a stable carbon layer, which can prevent the polyester material from further combustion.

Since the processing is often performed in air, the thermal decomposition ability of the polyester material in air atmosphere was further analyzed. A two-step thermal degradation behavior can be clearly observed. The first degradation step was similar to degradation in a N_2_ atmosphere. The *T*_5%_ and *T*_d, max_ in air were slightly lower than that in N_2_, but they were above 350 °C. First step degradation is purely thermal degradation. The second stage is the thermal oxidation process, which occurs at higher temperatures, where both *T*_5%_ and *T*_d, max_ in an air atmosphere of PBSF-PEG were somewhat higher than that relative to the PBSF raw material. This further confirms that the introduction of PEG does not deteriorate the thermal stability of PBSF, but inhibits the self-degradability of the copolyester at high temperatures. Therefore, it can be concluded that the incorporation of PEG improves the thermal stability of PBSF. As the *T*_5%_ in air is above 350 °C, which is evidently higher than the processing temperature of PBSF-PEG (150–180 °C) and the daily use temperature, it can be concluded that the PBSF-PEG has good thermal stability.

#### 3.3.2. Wide-Angle X-ray Analysis (WAXD) Analysis of PBSF and PBSF-PEG

In order to obtain a deep understanding the relationship between the structure and properties, it is important to study the effect of PEG units on the crystal structure of PBSF. The crystal structure of PBSF-PEG was explored by WAXD (Figure 6), and the peaks appeared at 2θ = 19.3°, 21.8°, 22.8° and 29.3°, which were similar to PBS [38]. It was found that the introduction of PEG did not change the structure of the PBSF crystals, probably because the test room temperature was close to or higher than the melting point of the PEG crystals, and thus the PEG crystals melted at the test temperature. Among them, the peaks’ intensity decreased, implying that the incorporation of the PEG unit decreased the copolymer crystallinity, which is also consistent with the DSC results.

#### 3.3.3. Water Contact Angle (WCA)

Hydrophilicity is an important property for fiber and their tissue engineering applications as hydrophilicity promotes cell adhesion and proliferation. The presence of bifunctional surfaces is very important for tissue engineering. Figure 7a represents the variation in the contact angles of the water drop onto the surface of the copolymers. After the introduction of the PEG soft segment, PBSF-PEG forms a block structure in which the hydrophilic and hydrophobic groups coexist, and the molecular chain of PEG is arranged loosely, so water molecules can easily stand on the surface of the PBSF-PEG and the hydrophilicity of the copolymers increased.

The water absorption capacity of the PBSF-PEG material was further investigated, as shown in Figure 7b, where the samples were immersed in water for 72 h and the change in the material mass was recorded. The water absorption of the PBSF-PEG sample changed significantly with the variation in the PEG content. It can be observed that when the content of the PEG chain segment was 10%, the water absorption was 4.21%, and the absorption increased regularly to 10.69% as the PEG chain segment increased. This result indicates that the PEG chain segment can significantly enhance the hydrophilicity of the copolymers and the hydrophilicity of the copolymers can be adjusted by controlling the content of the PEG chain segment. Enhanced water absorption facilitates the entrance of water molecules into the internal part of the copolyester, thus increasing the binding of water molecules with ester bonds and promotes the hydrolysis degradation.

#### 3.3.4. Degradation Properties of PBSF and PBSF-PEG

Biodegradable materials attract increasing interest as the white pollution caused by traditional nondegradable polymers [39]. Biodegradable PBSF-PEG has been synthesized and the degradation properties have been studied. Polyether chain segment has been introduced to regulate the degradation time. As discussed before, the hydrophilic PEG chain segments can improve the hydrophilic properties of polyester materials, which makes it easier for water molecules to enter into the interior of the polymer and attack the ester bonds, thus increasing the rate of ester bond breakage and the degradation of copolyesters. Meanwhile, water environments with different pH values affect the hydrolysis behavior of copolymers. The degradation behavior of PBSF-PEG was tested in PBS solution at pH = 6.84 and 0.01 mol/L NaOH solution at pH = 12, respectively. As shown in Figure 8a, the copolymers showed an increasing mass loss rate with increasing PEG content after the introduction of the PEG chain segments at pH = 6.84. When the PEG content was 30%, the residual mass was found to be 62%, while the residual mass of copolymers without the PEG segment was 81% after 24 days. Furthermore, it was evident that the size of the PBSF film after degradation was significantly larger that of PBSF-PEG after the introduction of the PEG segment (Figure 9).

The degradation of PBSF-PEG was significantly accelerated in NaOH solution (pH = 12), and all of the polyester films reached a high level of degradation at this pH after 15 days. The remaining mass of PBSF was 65% at 24 days. However, with the introduction of PEG, the degradation rate of the copolymers increased significantly, especially for the copolymers based on 30% of the PEG chain, where only 25% of the copolymer film remained. A significant decrease in the remaining size with increasing PEG content could also be clearly observed in Figure 9a. These results suggest that the degradation rate can be improved by the introduction of a polyether segment into the polyester material. [40]

#### 3.3.5. Mechanical Properties of PBSF and PBSF-PEG

Mechanical properties are significant for applications of material. Figure 10a shows the stress–strain curves of the copolymers before and after water absorption, and the mechanical parameters are summarized in Table 4. It could be observed from Figure 10 that PBSF, PBSF-PEG_10%_, and PBSF-PEG_20%_ presents three main stages: a linear increase of stress with strain, the appearance of a thin neck of yield, and finally a stress-hardening stage. However, when the introduced PEG content was above 20%, the yielding and necking became less evident, as seen in Figure 10. The yielding strength, tensile strength, and elastic modulus decreased with the increasing PBS content. It is well-known that the elastic modulus of polymers is closely related to their crystallinity and the content of flexible and rigid segments. With the increase in the content of the PEG segment, the crystallinity of thee copolymers decreases rapidly, which resulted in a significant reduction in the elastic modulus of the copolymers (Table 4). Meanwhile, the elongation at break increased with the introduction of flexible PEG segments, and the elongation at break reached as high as about 900% when the introduced PEG was 20%. The elongation at break of the copolymer reached over 1500% when the PEG content was 30%. This suggests that the introduction of flexible chain segments can significantly enhance the toughness of the modified copolymers.

In order to further study the effect of water absorption, the mechanical properties of the PBSF-PEG samples after water absorption were tested. It was found that the effect of water absorption on the mechanical properties was negligible when the PEG content was below 30%. However, when the introduced PEG content was 30%, the elongation at break of the copolymers decreased evidently, but other mechanical properties were almost unaffected. This decrease only occurred when the water absorption content was 10% or more. This may be due to the plasticization of PEG [41]. Therefore, we reach a safe conclusion that the PBSF-PEG copolymers can still maintain good mechanical properties after water absorption.

#### 3.3.6. Transmittance Properties of PBSF-PEG

It is widely recognized that the film’s UV resistance is essential to protect items from UV radiation, and thus is an important feature of food packaging films [42]. UV can be subdivided into UVA (320~400 nm), UVB (280~320 nm), UVC (200~280 nm), and visible light (450~800 nm) wavelengths [43]. The transmittance results are shown in Figure 11 and the average transmittance at 320–450 nm is listed in Table 5. The transmittance at 320 nm decreases significantly with the increase in the PEG content, which is mainly attributed to the introduction of PEG, leading to its intermolecular dispersion, resulting in the weakened absorption of UVC-UVB by the copolymers’ furan ring, and increased the transmittance performance.

Noteworthy, it can be found that the absorption in the visible region at 450 nm was significantly improved by the introduction of PEG, which is a great improvement for the application of copolymer films. Visible light transparency is an important characteristic as a food packaging film [44]. Although the introduction of PEG has some effect on UV absorption, PEG has the ability to improve the transparency in the visible region, which is beneficial for use in the food packaging field.

## 4. Conclusions

In this work, the effect of PEG on the thermal, mechanical, light transmission, and degradation properties of the PBSF-PEG copolymers was investigated. The introduction of an appropriate amount of PEG segments reduces the crystallization ability, but enhances the thermal stability and degradation rate of copolymers. The melting temperature of the copolymers of each component was above 100 °C, so they had good heat resistance performance. Regarding the mechanical property, the yield strength decreased from 24 MPa (PBSF) to 8 MPa (PBSF-PEG_30%_), while the elongation at break increased from 510% (PBSF) to 1500% (PBSF-PEG_30%_) with increasing PEG content. The copolymers possessed outstanding transparency, a broader processing window, and better mechanical properties compared to PBSF. In general, the higher the PEG content, the faster the hydrolysis rate. The degradation of copolymers can be further controlled by modulating the pH values of hydrolysis environments. Their hydrolysis rates can be easily adjusted accordingly, thus enabling farcical control of their degradation process, which is an effective attempt to further control the degradation process of the material using the control of the degradation environment.

## Figures and Tables

**Figure 1 polymers-14-04895-f001:**
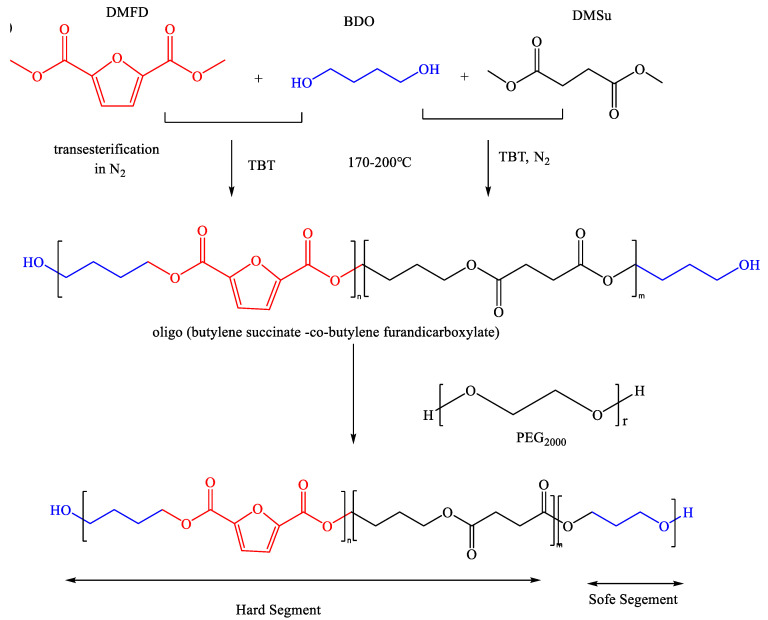
Synthesis process of the PBSF-PEG copolymers.

**Figure 2 polymers-14-04895-f002:**
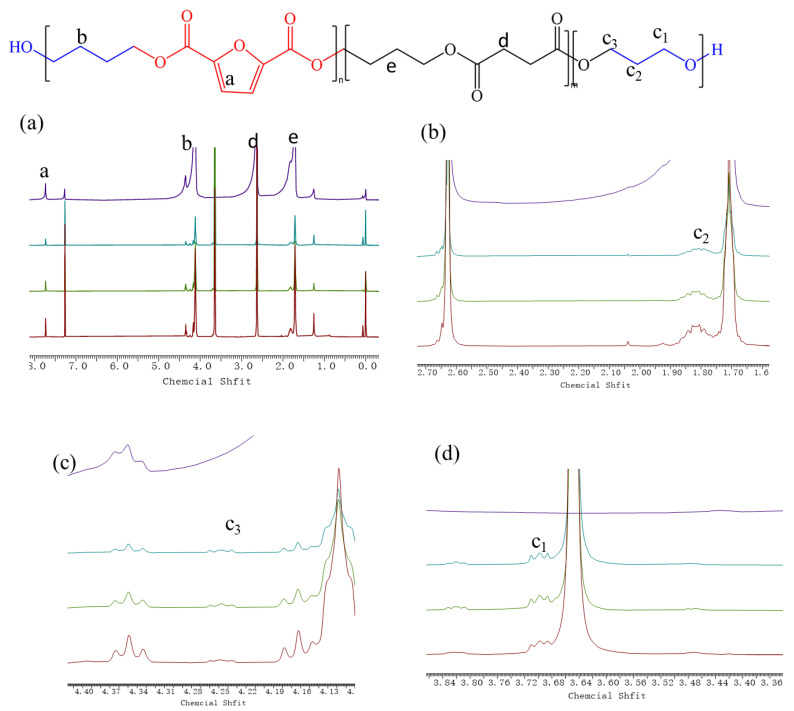
(**a**) ^1^H-NMR of the copolyesters and scheme 1. Chemical structures of PBSF-PEG (**b**–**d**) ^1^H-NMR of the copolyesters PEG.

**Figure 3 polymers-14-04895-f003:**
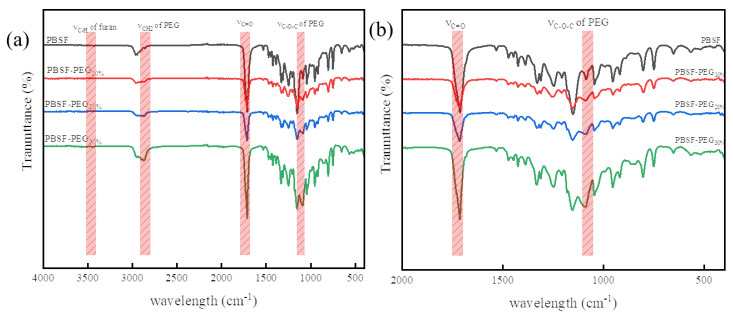
ATR-FTIR spectra of the copolymers (**a**) 4000–500 cm^−1^, (**b**) 2000–500 cm^−1^.

**Figure 4 polymers-14-04895-f004:**
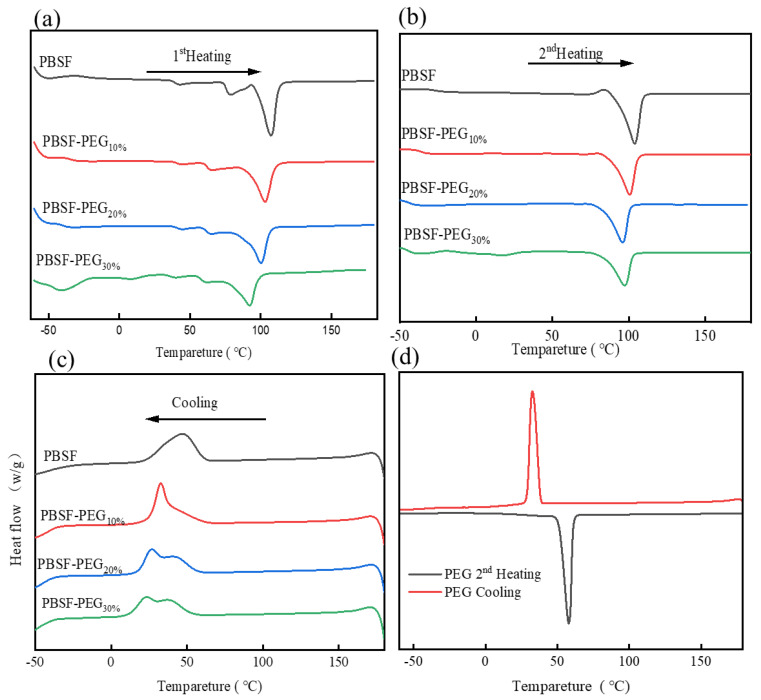
DSC curves of (**a**) first heating, (**b**) second heating, and (**c**) cooling. (**d**) PEG second heating and cooling.

**Figure 5 polymers-14-04895-f005:**
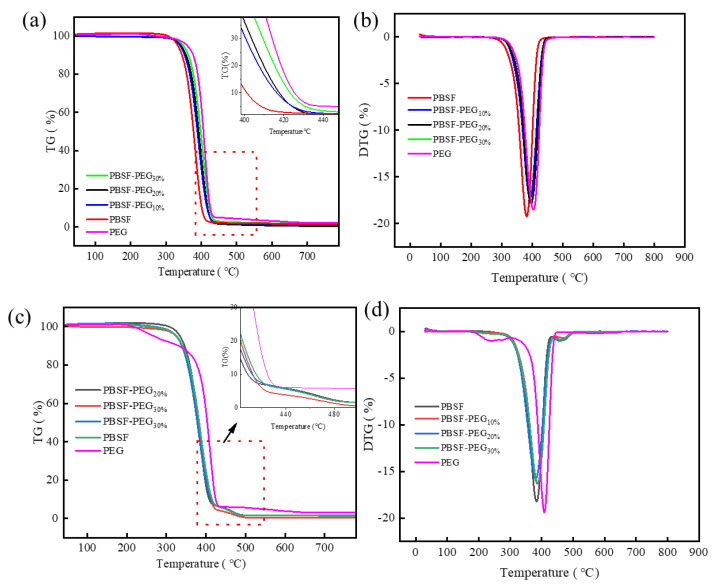
(**a**,**b**) TGA and DTG curves of PBSF, PBSF-PEG, and PEG in N_2_. (**c**,**d**) TGA and DTG curves of PBSF, PBSF-PEG, and PEG in air.

**Figure 6 polymers-14-04895-f006:**
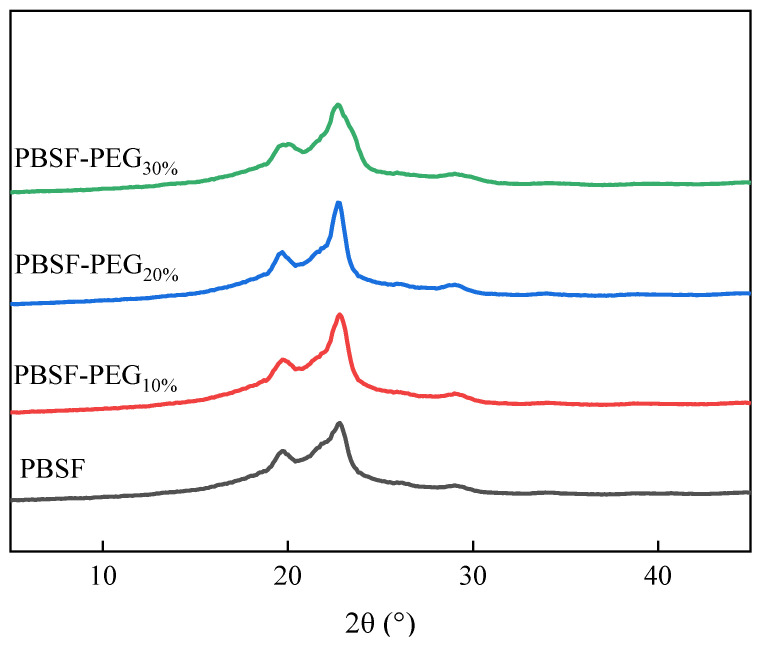
The WAXD pattern of PBSF and PBSF-PEG.

**Figure 7 polymers-14-04895-f007:**
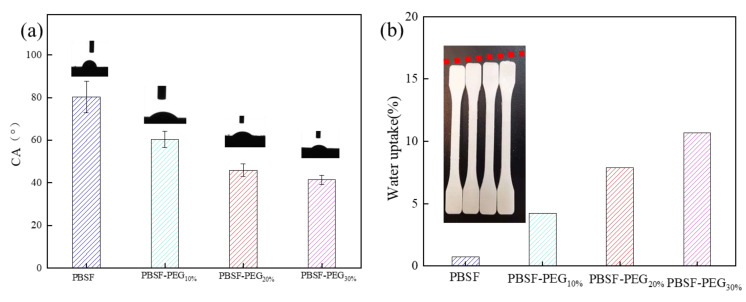
(**a**) Water contact angle (WCA) for the copolymer series with PEG 2000 and (**b**) water uptake percentage with time of PBSF-PEG.

**Figure 8 polymers-14-04895-f008:**
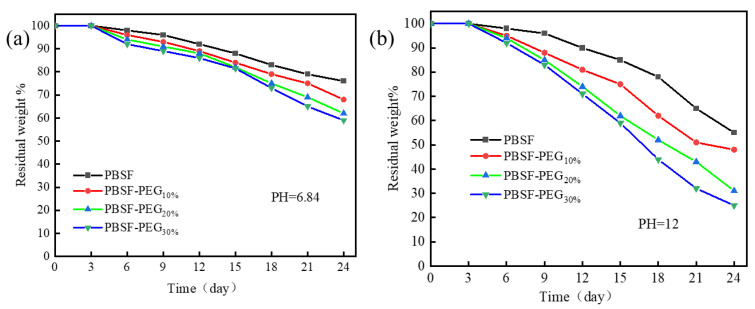
Residual weight as a function of degradation time in (**a**) PBS (pH = 6.84) and (**b**) NaOH solution (pH = 12) at 37 °C for PBSF-PEG.

**Figure 9 polymers-14-04895-f009:**
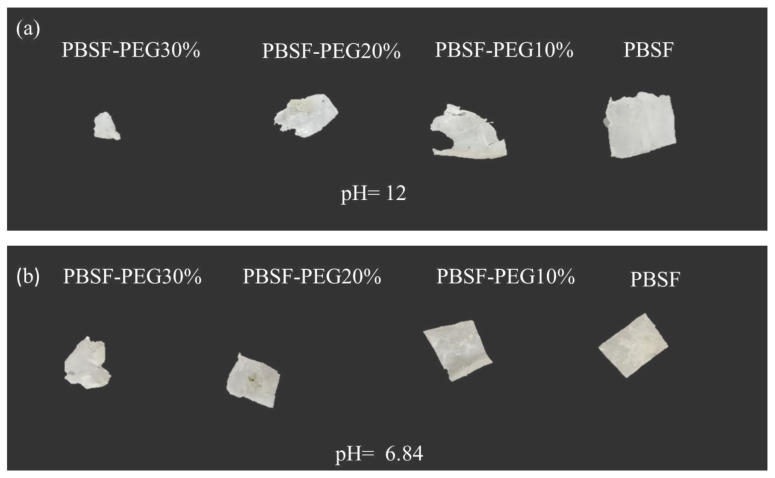
Photographs of the PBSF-PEG films after (**a**,**b**) hydrolytic degradation in NaOH solution and PBS solution.

**Figure 10 polymers-14-04895-f010:**
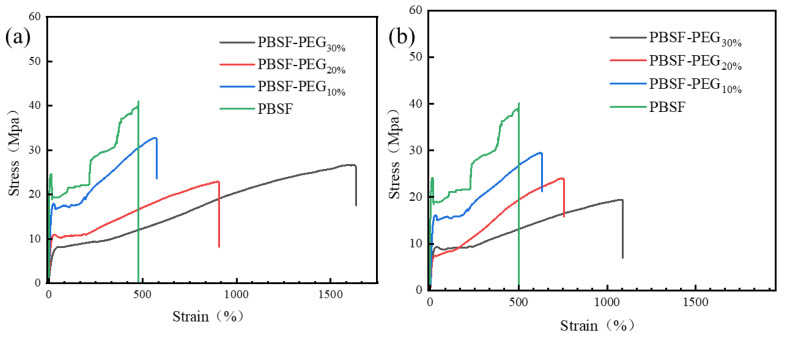
The representative stress–strain characteristics for the dry (**a**) and swollen (**b**) PBSF and PBSF-PEG.

**Figure 11 polymers-14-04895-f011:**
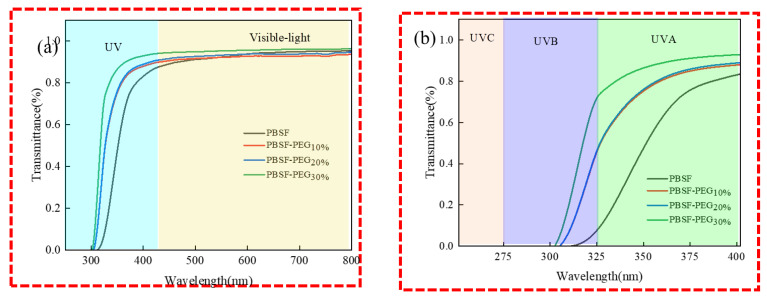
PBSF and PBSF-PEG film. (**a**) UV−Vis transmittance curves 800–200 nm. (**b**) UV−Vis transmittance curves 400–200 nm.

**Table 1 polymers-14-04895-t001:** Molecular structures, reaction condition, molecular weight of copolyesters, and PEG content calculated from the ^1^H NMR integral values.

Sample	PEG (mol%)		Polycondensation Temperature (°C)	[*η*]	*M*_n_ × 10^4^ (g/mol)	*M*_w_ × 10^4^ (g/mol)	PDI
	Feed	Found					
PBSF_10%_	nd	nd	230	1.65	5.48	10.81	1.96
PBSF-PEG_10%_	10.00	9.52	230	2.03	6.88	12.19	1.77
PBSF-PEG_20%_	20.00	19.04	230	2.25	7.90	13.68	1.73
PBSF-PEG_30%_	30.00	28.32	230	1.82	6.05	12.16	2.00
* PBS-PEG_10%_	--	--	--	--	6.32	8.59	1.36
* PBS-PEG_15%_	--	--	--	--	3.41	6.75	1.98

* PBS-PEG_10%_, * PBS-PEG_15%_ [14]. Note: -- represented “not detected”.

**Table 2 polymers-14-04895-t002:** Thermal properties of PBSF and PBSF-PEG.

Sample	DSC										WAXD
	1st Heating Scan		Cooling Scan		2nd Heating Scan						
	*T*_m_ (°C)	*ΔH*_m_ (J/g)	*T*_c_ (°C)	*ΔH*_c_ (J/g)	*T*_g_ (°C)	*T*_cc_ (°C)	*ΔH*cc (J/g)	*T*_m_ (°C)	*ΔH*_m_ (J/g)	*X*_c,h_ (%)	*X*_c_ (%)
PBSF	107.4	40.52	44.6	55.27	−26.6	84.5	5.78	105.2	45.89	37.5	29.5
PBSF-PEG_10%_	103.2	35.31	32.7	50.24	−35.3	80.2	1.37	100.7	41.41	33.4	27.8
PBSF-PEG_20%_	100.2	34.07	27.2	46.76	−41.1	64.6	3.08	98.5	43.15.	31.5	25.1
PBSF-PEG_30%_	98.3	25.92	23.5	43.81	−47.9	44.1	0.25	96.8	40.49	25.4	22.6
PEG	--	--	32.6	194.8	--	--	--	57.9	195.4	--	--

Note: -- represented “not detected”.

**Table 3 polymers-14-04895-t003:** Thermal stability of PBSF and PBSF-PEG.

Sample	TGA					
	N_2_			Air		
	*T*_5%_ (°C)	*T*_d,max_ (°C)	*R*_600_ (%)	*T*_5%_ (°C)	*T*_d,max_ (°C)	*R*_600_ (%)
PBSF	358.1	389.1	1.51	352.1	382.0	4.51
PBSF-PEG_10%_	360.4	393.2	1.10	355.8	391.7	3.56
PBSF-PEG_20%_	365.2	396.5	0.49	354.2	387.7	4.91
PBSF-PEG_30%_	370.0	404.1	1.31	359.8	393.9	4.57
PEG	265.5	409.3	2.24	243.5	408.4	3.23

**Table 4 polymers-14-04895-t004:** Mechanical characterization date of the dry and swollen PBSF and PBSF-PEG copolymers.

Sample	E (MPa)		$\delta_{y}$ (MPa)		$\delta_{b}$ (MPa)		*ε*_b_ (%)	
	Dry	Swollen	Dry	Swollen	Dry	Swollen	Dry	Swollen
PBSF	290 ± 20	290 ± 25	24 ± 2	24 ± 2	39 ± 3	39 ± 2	490 ± 30	510 ± 30
PBSF-PEG_10%_	249 ± 40	249 ± 40	18 ± 2	18 ± 1	31 ± 4	31 ± 3	520 ± 40	525 ± 40
PBSF-PEG_20%_	200 ± 20	198 ± 20	12 ± 2	12 ± 4	25 ± 3	25 ± 3	900 ± 30	915 ± 30
PBSF-PEG_30%_	178 ± 50	175 ± 50	8 ± 1	8 ± 2	23 ± 1	23 ± 1	1580 ± 40	1240 ± 40

**Table 5 polymers-14-04895-t005:** UV–Vis transmittance values at maximum UV-blocking wavelength of the polyester films.

Sample	*T*_320_ (%)	*T*_380_ (%)	*T*_400_ (%)	*T*_450_ (%)
PBSF	3.2	47.5	83.3	89.1
PBSF-PEG_10%_	32.4	75.9	88.7	90.6
PBCBS-PEG_20%_	58.9	76.6	89.6	91.5
PBCBS-PEG_30%_	58.9	86.5	93.2	95.4

## Data Availability

Not applicable.

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
