# Peer review of "Synthesis of Biodegradable Polyester–Polyether with Enhanced Hydrophilicity, Thermal Stability, Toughness, and Degradation Rate"

_polymers, 2022, doi:10.3390/polym14224895_

Round 1
Reviewer 1 Report
This paper deals with the synthesis and characterization of poy (butylene succinate-butylen furandycarboxilate) (PSBF) / poly (ethylene glycol) (PEG) copolymers. Copolymers of high molecular weight were synthesized and the molecular parameters by GPC and NMR successfully provided. The effect of PEG amounts up to 30% to PSBF on hydrophobicity and thermal and mechanical properties is studied. The investigation is sound and the system may be of interest for a polymer science community. However, a more insightful discussion of results is required. The mechanical properties section is particularly faulty and must be rewritten carefully. Grammar and syntax must be carefully checked all over the manuscript; mistakes are common i.e: lines 122, 123, 173-174, 203, 276, Fig.1 caption etc.
Author Response
Response to Reviewer 1 Comments
To Reviewer 1: This paper deals with the synthesis and characterization of poy (butylene succinate-butylen furandycarboxilate) (PSBF) / poly (ethylene glycol) (PEG) copolymers. Copolymers of high molecular weight were synthesized and the molecular parameters by GPC and NMR successfully provided. The effect of PEG amounts up to 30% to PSBF on hydrophobicity and thermal and mechanical properties is studied. The investigation is sound and the system may be of interest for a polymer science community. However, a more insightful discussion of results is required. The mechanical properties section is particularly faulty and must be rewritten carefully. Grammar and syntax must be carefully checked all over the manuscript; mistakes are common i.e: lines 122, 123, 173-174, 203, 276, Fig.1 caption etc.
Response 1: Thanks very much for your comments, which are very helpful to improve the quality of this article. For the problem of mechanical properties. We have checked the manuscript carefully and thoroughly and correct all the grammar and typos. For example, line 352-353 have been revised as, ‘It can be observed from Figure 10 that PBSF, PBSF-PEG10% and PBSF-PEG20% presents three main stages: linear increase of stress with strain, appearance of a thin neck of yield, and finally a strain-hardening stage.’, lines 355-357 has been corrected as ‘The yielding strength, tensile strength and elastic modulus decreases with increasing PBS content.’ lines 369-371 has been replaced with ‘However, when the introduced PEG content is 30%, the elongation at break of copolymers decrease evidently, while other mechanical properties were almost unaffected.’ and lines 372-374 have been revised as, ‘it may be due to the plasticization of PEG.41 Therefore, it can reach a safe conclusion that the PBSF-PEG copolymers can still maintain good mechanical properties after water absorption.’
Response 2: We have checked the whole manuscript very carefully and revise the article according to your suggestion; please see the manuscript or the following: lines 122,123:‘entropy’ has been revised as ‘entropy of cold crystallization’; lines 173-174 “arising from a different chemical environment of this methylene group” has been corrected as “arising from their different chemical environment”; line 203: ‘were’ has been revised as ‘are’; Line 275-276, “Hydrophilicity of the polymer is an important property for fiber and tissue engineering applications. Because hydrophilicity promotes cell adhesion and proliferation, which” has been replaced with “Hydrophilicity of the polymer is an important property for fiber and their tissue engineering applications, as hydrophilicity promotes cell adhesion and proliferation.”; Fig.1 caption “Synthesis of PBSF-PEG via two-step melt polycondensation of DMSu, BDO, DMFD and PEG.’ has been revised as ‘Figure 1. Synthesis process of PBSF-PEG copolymers.
Reviewer #2: In this manuscript, the authors synthesized a pegylated polyester based poly(butylene succinate-butylene furandicarboxylate/ polyethylene glycol succinate) (PBSF-PEG). Multiple characterizations and systematical investigations have been conducted. I think the analysis is detailed, but there are some concerns about experimental design and result discussion. I would like to recommend a minor revision before the manuscript can be accepted.

Reviewer 2 Report
In this manuscript, the authors synthesized a pegylated polyester based poly(butylene succinate-butylene furandicarboxylate/ polyethylene glycol succinate) (PBSF-PEG). Multiple characterizations and systematical investigations have been conducted. I think the analysis is detailed, but there are some concerns about experimental design and result discussion. I would like to recommend a minor revision before the manuscript can be accepted.
Comments:
1. Introduction: The authors should give a broader vision of degradable polyesters and PEGylated polyesters instead of being limited to PBS or PBSF. For example:PGS, PCL PGA and their PEGylated derivatives (Polyesters: 10.1016/j.jconrel.2022.01.035; doi.org/10.1016/j.eurpolymj.2021.110830;etc. PEGylated polyesters: doi.org/10.1016/j.msec.2019.01.057; doi.org/10.1016/j.actbio.2016.08.023; doi.org/10.1007/s40005-019-00442-2; doi.org/10.3390/polym11060965; etc.)Please refer to more recent works or reviews to convince readers that such investigation can also benefit other polyester modifications.
2. How did the authors choose the Mw of PEG? Why was 2000 used instead of other Mw? Is 2000 PEG the most environmentally friendly?
3. Please also provide pure PGE in TG and DSC for comparison.
4. Since the potential applications are focused on sustainable plastic packaging materials, the degradation performance should be performed in a soil environment or similar microbial environment. The water environment is more likely to test its degradation under physiological conditions.
5. How about the safety assessment of the pegylated polyesters? Will any toxicity be included if they are used for food wrapping?
Author Response
Response to Reviewer 2 Comments
General comments
- Introduction: The authors should give a broader vision of degradable polyesters and PEGylated polyesters instead of being limited to PBS or PBSF. For example:PGS, PCL PGA and their PEGylated derivatives (Polyesters: 10.1016/j.jconrel.2022.01.035; doi.org/10.1016/j.eurpolymj.2021.110830;etc. PEGylated polyesters: doi.org/10.1016/j.msec.2019.01.057; doi.org/10.1016/j.actbio.2016.08.023; doi.org/10.1007/s40005-019-00442-2; doi.org/10.3390/polym11060965; etc.)Please refer to more recent works or reviews to convince readers that such investigation can also benefit other polyester modifications.
Response: According to the suggestion of the reviewer, we review more recent works and add more references to convince readers that our investigation can also benefit other polyester modifications; please see page 4 or the following:
In recent years, potential applications of biodegradable aliphatic polyesters and copolyesters, such as poly(glycerol sebacate)17, poly(ε-caprolactone)18, poly(glycolic acid) 19 and their PEGylated derivatives19-22.
- Feng, X.; Wang, G.; Neumann, K.; Yao, W.; Ding, L.; Li, S.; Sheng, Y.; Jiang, Y.; Bradley, M.; Zhang, R., Synthesis and characterization of biodegradable poly(ether-ester) urethane acrylates for controlled drug release. Materials Science and Engineering: C 2017, 74, 270-278.
- Sha, D.; Wu, Z.; Zhang, J.; Ma, Y.; Yang, Z.; Yuan, Y., Development of modified and multifunctional poly(glycerol sebacate) (PGS)-based biomaterials for biomedical applications. European Polymer Journal 2021, 161, 110830.
- Dethe, M. R.; A, P.; Ahmed, H.; Agrawal, M.; Roy, U.; Alexander, A., PCL-PEG copolymer based injectable thermosensitive hydrogels. Journal of Controlled Release 2022, 343, 217-236.
- Perinelli, D. R.; Cespi, M.; Bonacucina, G.; Palmieri, G. F., PEGylated polylactide (PLA) and poly (lactic-co-glycolic acid) (PLGA) copolymers for the design of drug delivery systems. Journal of Pharmaceutical Investigation 2019, 49 (4), 443-458.
- Wang, J.-Z.; You, M.-L.; Ding, Z.-Q.; Ye, W.-B., A review of emerging bone tissue engineering via PEG conjugated biodegradable amphiphilic copolymers. Materials Science and Engineering: C 2019, 97, 1021-1035.
- Ma, Y.; Zhang, W.; Wang, Z.; Wang, Z.; Xie, Q.; Niu, H.; Guo, H.; Yuan, Y.; Liu, C., PEGylated poly(glycerol sebacate)-modified calcium phosphate scaffolds with desirable mechanical behavior and enhanced osteogenic capacity. Acta Biomaterialia 2016, 44, 110-124.
- Wang, Y.; Wu, H.; Wang, Z.; Zhang, J.; Zhu, J.; Ma, Y.; Yang, Z.; Yuan, Y. Optimized Synthesis of Biodegradable Elastomer PEGylated Poly(glycerol sebacate) and Their Biomedical Application Polymers [Online], 2019.
- How did the authors choose the Mw of PEG? Why was 2000 used instead of another Mw? Is 2000 PEG the most environmentally friendly?
Response: We chose PEG with a Mw of 2000 instead of another Mw because as compared to copolymers synthesized from shorter PEG segments, the copolymer based on longer PEG segment (a molecular weight of 2000 g/mol) exhibit higher swelling degree in water and faster degradation rate. Due to the excellent hydrophilicity and biocompatibility, PEG has been approved by the U.S. Food and Drug Administration (FDA) for a variety of medical applications.1-2 Therefore, PEG is regarded to be most environmentally friendly among the polymers.
- Kwizera, E. A.; Ou, W.; Lee, S.; Stewart, S.; Shamul, J. G.; Xu, J.; Tait, N.; Tkaczuk, K. H. R.; He, X., Greatly enhanced CTC culture enabled by capturing CTC heterogeneity using a PEGylated PDMS-titanium-gold electromicrofluidic device with glutathione-controlled gentle cell release. ACS Nano 2022, 16 (7), 11374-11391.
- Liu, K.; He, Y.; Yao, Y.; Zhang, Y.; Cai, Z.; Ru, J.; Zhang, X.; Jin, X.; Xu, M.; Li, Y.; Ma, Q.; Gao, J.; Lu, F., Methoxy polyethylene glycol modification promotes adipogenesis by inducing the production of regulatory T cells in xenogeneic acellular adipose matrix. Materials Today Bio 2021, 12, 100161.
- Please also provide pure PGE in TG and DSC for comparison.
Response: We have provided PGE in TG and DSC for comparison and some corresponding comments have been made; please see Figure 4 and Figure 5 or the following:
Figure 4 DSC curves of (a) 1st heating, (b) 2nd heating and (c) cooling (d) PEG 2nd heating and cooling
Figure 5. (a,b) TGA and DTG curves of PBSF, PBSF-PEG and PEG in N2, (c,d) TGA and DTG curves of PBSF, PBSF-PEG and PEG in air
The thermal properties of PBSF, PBSF-PEG and PEG were characterized by DSC (Fig. 4) and TGA (Fig. 5). For PEG2000, there is a cold crystallization peak (32.6 °C) and a melting peak (57.9 °C) in the second heating curve in Figure 4(d). However, no cold crystallization peaks and melting peaks of PEG segment can be observed in PBSF-PEG and PEG.
It can be found that T5% of copolymers by about 10 °C after the introduction of PEG, though the initial decomposition temperature of PEG T,5d% is around 265.5 °C, which is much lower than the initial decomposition temperature of PBSF-PEG is ranging from 358 to 370 °C.
- Since the potential applications are focused on sustainable plastic packaging materials, the degradation performance should be performed in a soil environment or similar microbial environment. The water environment is more likely to test its degradation under physiological conditions.
Response: We are deeply grateful for this professional suggestion; the degradation of polymers in soil and similar microbial environments is known to be critical, but there is no way to conduct this experiment is time consuming as it needs 3 months or longer time, and there is only 10 days for us to revise and return the manuscript. We will study the degradation behavior in soil degradation and similar microbial environments in a subsequent study.

Round 2
Reviewer 1 Report
Section 3.4.5 Mechanical properties still contains many misspellings that must be corrected.
Section 1. Introduction still contains some misspellings.
The Conclussions section can be improved.
Author Response
Response to Reviewer 1 Comments
Section 3.4.5 Mechanical properties still contains many misspellings that must be corrected.
Response 1: Thanks very much for your comments, which are very helpful to improve the quality of this article. For the problem of mechanical properties. We have checked the manuscript carefully and thoroughly and correct all the grammar and spellings. For example; lines 352 ‘’street-stain’’ has been revised as ‘’stress–strain’’, lines 355 ‘’presents’’ has been revised ‘’present’’, lines ‘’decreases and PBS’’ have been revised as ‘’decrease and PEG’’, lines 370 ‘’can be’’ have been revised as ‘’could be’’.
Section 1. Introduction still contains some misspellings.
Response 2: We have checked the whole manuscript very carefully and revise the introduction according to your suggestion; please see the introduction or the following: For example; lines 46 ‘’to’’ has been corrected as ‘’and’’, lines 54 ‘’exhibit’’ has been corrected as ‘’exhibits’’, lines 61 ‘’varying content’’ has been revised ‘’varying contents’’, lines 72 ’’tend’’ has been corrected as ‘’tends’’, lines 75 ‘’ show’’ has been revised as ‘’showed’’.
The Conclusions section can be improved.
Response 3: According to the suggestion of the reviewer, we add more details in the conclusion to convince readers that our investigation can also benefit other polyester modifications; please see page 24 or the following:
As for the mechanical property, the yield strength decreased from 24 MPa (PBSF) to 8 MPa (PBSF-PEG30%), while the elongation at break increased from 510% (PBSF) to 1500% (PBSF-PEG30%). The copolymers possessed outstanding transparency, broader processing window and better mechanical properties as compare to PBSF.
